# Thoracic Biometry in Patients with Congenital Diaphragmatic Hernia, a Magnetic Resonance Imaging Study

**DOI:** 10.3390/diagnostics14060641

**Published:** 2024-03-18

**Authors:** Erick George Neștianu, Septimiu Popescu, Dragoș Ovidiu Alexandru, Laura Giurcăneanu, Radu Vlădăreanu

**Affiliations:** 1Faculty of Medicine, “Carol Davila” University of Medicine and Pharmacy, 030167 Bucharest, Romania; 2Hôpital Bicêtre AP-HP, 94270 Le Kremlin-Bicêtre, France; 3Faculty of Medicine, L’Université Paris-Saclay, 75015 Paris, France; 4Department of Medical Informatics and Bio-Statistics, University of Medicine and Pharmacy of Craiova, 200349 Craiova, Romania; 5University Hospital of Coventry and Warwickshire NHS Trust, Coventry CV2 2DX, UK; 6Department of Obstetrics and Gynecology, Elias University Emergency Hospital, 011461 Bucharest, Romania

**Keywords:** magnetic resonance imaging, ultrasound, congenital diaphragmatic hernia, lung to head ratio, thoracic biometry, pulmonary hypoplasia

## Abstract

This is a retrospective study investigating biometric measurements using magnetic resonance imaging (MRI) examinations in congenital diaphragmatic hernia (CDH). CDH is one of the more common causes of pulmonary hypoplasia, with grave consequences for the fetus. Inclusion criteria were patients diagnosed with CDH as the only observed anomaly, who underwent MRI examination after the second-trimester morphology ultrasound. The patients came from three university hospitals in Bucharest, Romania. In total, 19 patients were included in the study after applying exclusion criteria. Comparing the observed values of the thoracic transverse diameter, the thoracic anterior–posterior diameter, the thoracic circumference, the thoracic area, and the thoracic volume with values from the literature, we observed a predictive alteration of these parameters, with most showing Gaussian distribution. We observed statistical significance for most of our correlations, except between the observed and expected thoracic anterior–posterior diameters and the observed and expected thoracic volume values. This is very helpful when complex studies that can calculate the pulmonary volume cannot be obtained, as in the case of movement artifacts, and allows the clinicians to better assess the severity of the disease. MRI follow-up in CDH cases is a necessity, as it offers the most accurate thoracic biometry.

## 1. Introduction

Before talking about congenital diaphragmatic hernia (CDH), first we must pass over what makes it such a difficult disease to manage, that being the apparition of pulmonary hypoplasia.

Pulmonary hypoplasia is characterized by an abnormal, incomplete development of the fetal lung with various consequences for the fetal development. This might lead to various levels of respiratory insufficiency after birth [1].

It affects about 1.4% of total live births, although the real prevalence is not known. The Fetal Medicine Foundation considers the prevalence to be about one in fifty thousand births [2,3].

The origin of pulmonary hypoplasia is the incomplete development of pulmonary tissue. This phenomenon can happen unilaterally or bilaterally. Clinically, respiratory failure might be present depending on the missing pulmonary volume, the number of remaining alveoli, and the correct development of the bronchial ramifications. Severity is heavily dependent on the moment the injuring factor appears during embryo development. Early lesions will cause more damage and increase the severity [4].

We know of multiple intrathoracic (extralobar sequestration, diaphragm agenesis, mediastinal masses or tumors) or extrathoracic (oligoamnios, preterm premature rupture of membranes, skeletal dysplasia, large intraabdominal masses, and neuromuscular conditions) causes that determine pulmonary hypoplasia, with different pathophysiological pathways [5,6,7].

Mechanical stimuli are also an important part of lung development. They appear at around ten weeks of gestation, providing needed stimulation to the epithelial cells by allowing the movement of fluids through the airways [8].

The leading cause of pulmonary hypoplasia is congenital diaphragmatic hernia, followed by pulmonary sequestration, mediastinal masses (teratoma, thymoma), and perfusion anomalies [9]. The most vulnerable to negative influences for the lung is the pseudo-glandular phase when we see a rapid development of vascular structures and the bronchial airways [2].

The first line in detecting CDH is the first-trimester ultrasound. In addition to observing the diaphragmatic defect and/or herniated organs in the thoracic cavity, a few standardized measurements are taken. The lung-to-head ratio (LHR) is a well-established measurement but one that is unreliable, tending to underestimate the severity of lung hypoplasia. In addition to this, we can also look at thoracic biometric values of the anterior and transverse diameters, thoracic area, and thoracic circumference.

Another element that we come to observe is a major mediastinal shift. As such, the position and size of the heart might also provide insight when assessing these cases. Although multiple 2D measurements can be made, the best assessment method is calculating the lung volume [10].

3D sonography is not used for routine lung volumetry because it is too time-consuming and might not be an accurate predictive tool. Using multiplanar mode allows the simultaneous display of three perpendicular views of the lung, thus obtaining its volume [10].

For a rigorous and accurate severity assessment, the patient should have a high-resolution ultrasound, with an eventual virtual organ computer-aided analysis (VOCAL) study, magnetic resonance imaging (MRI), echocardiography, and genetic testing between 20 and 24 weeks gestation. This leaves sufficient time to terminate the pregnancy if needed and also to offer rigorous parental counseling envisioning the possible treatments and management but also the possible complications and outcomes [11,12].

Fetal MRI is already an established and frequently used method to further explore lung hypoplasia in the case of CDH, providing excellent soft tissue contrast and high spatial resolution with a large field of view [10,13].

In the case of CDH, studies have shown that MRI is a better survival predictor, followed by the 3D and 2D methods, although the lung-to-head ratio (LHR) method is not to be ignored, still showing its utility in many cases [14].

In this study, we aim to look at the correlation between 2D and 3D volume biometric values obtained by MRI and ultrasound to see if other important features can be used in severity assessment other than the direct measurement of total pulmonary volume. The importance of correct diagnosis of pulmonary hypoplasia is due to the special treatment needed for these patients. They should be surveyed in specialized institutions with trained and experienced teams [15].

## 2. Materials and Methods

This is a multicenter retrospective study. Data were collected between January 2019 and December 2022 from three university hospitals in Bucharest, Romania, specializing in antenatal diagnosis and treatment.

We included patients who underwent second-trimester fetal morphology screening in these hospitals. Pregnancy had to occur naturally, without in vitro fertilization, and CDH had to be the only observable anomaly in these patients. All patients signed a formal consent form before entering the study. Approval from the hospitals’ Ethical Committee was also obtained before the beginning of the study in December 2017, protocol number 3180.Patient follow-up was performed as per ISUOG guidelines for the second and third trimesters.

The inclusion criteria were as follows: naturally pregnant women (not by in vitro fertilization) who had undergone the second-trimester fetal morphology ultrasound in specialized diagnosis centers that had discovered a CDH as the only observable malformation and who had had a follow-up MRI examination to confirm or complete the ultrasound diagnosis.

The ultrasound examination was performed on devices dedicated to obstetrical and fetal morphology analysis using specialized transducers.

For the MRI examination, we used 1.5 Tesla machines, as current guidelines dictate, with the aid of body coils. The usual sequences used were Fast Imaging Employing Steady State Acquisition (FIESTA), Single Shot Fast Spin Echo (SSFSE), Liver Acquisition with Volume Acceleration (LAVA), and occasionally diffusion-weighted Weighted Image (DWI). The slice thickness was between 2 and 6 mm depending on the presence or absence of movement artifacts. The investigation was made with the mother in a supine or lateral position, using sedation (premedication with 7.5 mg of Zopiclone) in most cases.

Fetal lung volume was calculated by using a new method to reduce the overestimation of the lung volume. The first step was tracing the lung area of both lungs on each slice. We then obtained the mean between every 2 consecutive slices and multiplied the value by the distance between them. Finally, we calculated the sum of all these values and obtained the TLV for each lung. For easier viewing of this parameter, we calculated the total lung volume ratio (TLVR) by calculating the ratio between our observed value and the expected lung volume values obtained from reference articles that measured standard values for lung volumetry using MRI. To be more precise, we looked at the studies of Meyers et al., Rypens et al., Osada et al., and Sefidbakht et al. [16,17,18,19] and calculated a mean from them that was used as our reference. Figure 1 and Figure 2.

We also compared our findings with the expected values obtained by using ultrasound volumetry as per the work of Lian et al. [20].

For a better visualization of these values, we calculated the total lung volume MRI index by dividing the observed value of the lung volume by the expected value calculated with the aforementioned method. We also made the same calculation using expected pulmonary volume values from ultrasound studies [20].

Lung tracing was undertaken in general in the axial plane, but we also made use of the coronal or sagittal planes in some cases where the axial acquisition was artifacts by the movement of the fetus.

We also measured fetal biometrics such as the transverse and anterior–posterior diameters of the thoracic cavity. To further study the effect of the herniated organs on the thoracic cavity we also measured the thoracic area and perimeter following the same methods used by Xihua Lian et al. [20].

The thoracic transverse and anterior–posterior diameters were measured in the axial plane at a level equivalent to the ultrasound four-chamber section. The measurements were made from the internal face of the ribs for the transverse diameter and from the posterior face of the sternum to the vertebral body for the anterior–posterior diameter respectively. Figure 3 and Figure 4.

The thoracic circumference was measured following the outer margin of the ribs and respectively tracing the posterior contour of the vertebral transverse and spinous processes. Figure 5 and Figure 6.

The thoracic area, however, was measured on the interior margin of the ribs, the spinous processes, and the vertebral body. Figure 7 and Figure 8.

Furthermore, using the same method of calculating the lung volumes, we also measured the total thoracic cavity volume considering the upper margin of the thorax the superior thoracic aperture, and the lower margin of the diaphragm. This proved to be somewhat of a challenge when the herniation defect was larger, as the diaphragm was not so easily identified in some cases. Seeing as the diaphragm presented a loss of substance and continuity, we tried to identify the insertion of the diaphragm to the thoracic wall and trace an imaginary line that connected to its origin on the other side. This was best undertaken in sagittal images, but sometimes we found the coronal images to be better. Figure 9.

All of the measurements above were made by the same person, analyzing the MRI examinations that were anonymized beforehand. This was performed using the RadiAnt DICOM Viewer.

Data were recorded using Microsoft Excel files; statistical analysis was performed using MS Excel (Microsoft Corp., Redmond, WA, USA) and the XLSTAT add-on for MS Excel (Addinsoft SARL, Paris, France). Descriptive analysis of the study groups was performed with MS Excel. Statistical tests (Mann–Whitney test for the comparison of non-parametric numerical data, Pearson’s correlation test) were performed using the XLSTAT add-on.

## 3. Results

We enlisted a total of 23 patients in our study. A total of 19 fulfilled the inclusion criteria, as one patient got pregnant through in vitro fertilization and the other had associated posterior fossa malformations.

The main factor we wanted to study was how the thoracic biometry is altered in the presence of a diaphragmatic hernia.

As such we compared the observed values of the thoracic transverse diameter, the thoracic anterior–posterior diameter, the thoracic circumference, the thoracic area, and the thoracic volume with expected values from the literature [20].

Results showed alterations to all these parameters in an expected fashion, showing in general an increase in the thoracic diameters, the thoracic circumference, and the thoracic volume. Only two parameters were found to have lower-than-expected values: the total lung volume, which is to be expected in any congenital diaphragmatic hernia, and the thoracic area at the four-chambers level.

An interesting note is to be made by looking at the thoracic circumference and thoracic area. While it would seem that both of these values should have been larger than expected, especially after finding both thoracic diameters to be larger than the expected values, in reality, we found that only the thoracic circumference had larger-than-expected values, whilst the thoracic area had smaller-than-expected values.

As seen in the following table (Table 1), almost all of the studied parameters had a Gaussian distribution, with the only exceptions being the observed thoracic volume and the lung-to-thoracic volume ratio.

As such, we decided to use the parametric Student’s t-test for paired samples and found the following results, as shown in the second table (Table 2).

We observed statistical and high statistical significance for most of our correlations. The only ones that showed no statistical significance were between the observed and expected thoracic anterior–posterior diameters and the observed and expected thoracic volume values.

## 4. Discussion

Looking through the data, it becomes clear to see that normal thoracic biometry is affected in the fetuses with CDH. The most important parameter that we studied is the total lung volume, as it has the most negative repercussions in the later development of the child, in some cases being so severely lowered that it becomes incompatible with life.

As shown in previous studies, the lung-to-head ratio (LHR) is not able to correctly assess the severity of the hernia and is very imprecise when it comes to assessing the lost lung volume. As such, we wondered if we might be able to find other biometric parameters that can better correlate with the real volume loss and maybe offer a more thorough ultrasound assessment. We find this to be of important diagnostic value, as there are still many places where fetal MRI is not readily available for everyone in need, and as such a better ultrasound evaluation protocol might offer similar results by looking at the LHR and also other parameters [20].

As seen in the table above, most of the data we collected proved to be statistically significant, showing a Gaussian distribution. This might support its use in routine practice with most parameters.

Calculating only the LHR for a severity evaluation of CDH seems not to be enough, as the high variability of this measurement generally underestimates the actual volume loss. A solution might be not to assess severity using only the LHR when MRI or VOCAL studies are not available but also by looking at the biometric values of the thorax. We have shown that there are statistically relevant connections between the transverse diameter, thoracic area, circumference, and lung-to-thoracic-area ratio as well as the total pulmonary volume [10].

Most of the correlations we observed also proved to be statistically relevant, with the most interesting and useful one showing the fact that the VOCAL technique is as good as MRI when it comes to calculating the total pulmonary volume. This shows the great accuracy of both methods when it comes to calculating lung volume and helps the clinician when it comes to deciding the right assessment method for each patient. This is of great interest, seeing as in some places MRI is not readily available, and perhaps VOCAL exploration might be a good alternative if it is easier to access. As we already know, lung volume is the best way of assessing severity and also future respiratory morbidity [21].

In the case that VOCAL studies are unavailable, calculating the LHR is a good first step in trying to assess severity, but it might be recommended to also take a look at the other biometric values and compare them to standardized tables. Comparing them will offer a better understanding of the true severity, even though the best approach is to calculate the total lung volume. As shown in other specialized literature, we can still benefit from 2D analysis [22].

An aspect that we should take note of is the fact that the anterior–posterior diameter of the thoracic cavity presents no statistical correlation between the observed and expected values. Although it might seem logical that we should find this parameter augmented, it does not provide much in the sense of severity assessment and should not be a main factor when analyzing the severity.

In contrast, the transverse thoracic diameter proves to have a high statistical value when correlating the observed and expected values, making it much more valuable when analyzing fetal biometrics. The same also applies to the thoracic circumference and thoracic area. This might prove very useful in the normal medical practice of centers that do not have access to advanced ultrasound techniques, such as the VOCAL method or MRI.

The thoracic volume appears to not have any statistical correlation between the observed and expected values, although the *p*-value is 0.0518. There is a possibility that with a larger patient pool, this value might also achieve statistical relevancy, but further research is required.

Our data show high statistical relevancy when correlating the observed total pulmonary volume with expected values obtained from MRI studies as well as VOCAL studies. This means that we can compare the observed values with either the ultrasound or the MRI standardized values and obtain a satisfactory outcome.

Other studies searched for correlations between pulmonary hypoplasia and the biometric measurements that can be readily acquired during pregnancy. Some of them had promising results, although the recent literature shows low sensitivity and accuracy [23].

There is a linear correlation between thoracic size and gestational age when discussing a normal pregnancy. Normally, there is a constant ratio between thoracic size and other biometric indices such as the biparietal diameter, the head circumference, the abdominal circumference, and the femur length. The most well-known and also most used in practice is the LHR method, which takes into account the cranial circumference. They also offer an easily reproducible method of acquisition, requiring standard sections, such as the four-chamber view being the most commonly used among them when discussing thoracic pathology [24,25].

There are studies that instead of looking at the global severity of the disease, just resorted to determining if the volume loss would prove to be fatal or not, such as the case in the study of Vintzileos et al. [26]. While we find this to be a good step forward, we should try to find new methods of 2D and 3D examinations that can correctly and constantly assess severity. The study has looked at multiple biometric parameters in search of the best one to predict lethality. The most promising element shown in the study was a ratio calculated using the difference between the thoracic area and the heart area that was divided by the thoracic area. While we did not use this parameter in our study, it might be interesting to further explore the possibility in the future [27].

Other studies have shown that the thoracic circumference/abdominal circumference ratio is a good predictor for fetal lung hypoplasia. The problem with this parameter when discussing CDH is that we have already seen an alteration of thoracic parameters and can presume that the abdominal parameters might also be altered. This is an important point of view, as in the case of CDH the ratio might not be altered in a predictable pattern. In our study, we only took notice of thoracic biometric parameters, but in the future a study centered on the correlation between thoracic and abdominal biometrics might provide some interesting insight into the disease [28].

Recent studies have shown that even postpartum therapeutic intervention will result in significant volume recovery [29]. This means that correct and early severity assessment is more important than ever because not all cases might need prenatal interventions that come with risks for the mother and the fetus. In light cases where we observe little volume loss and a small probability of impact on the child after birth, the best course of action might be to wait for childbirth and perform surgery on the newborn. The recent literature has shown the ability to accurately measure and envision the diaphragmatic defect location as well as its dimensions and approximate shape using 3D rendering [30].

Although it was not within the scope of this article, our team has also started mapping the herniation defect and its size. This information is of great help to the surgeon, who will have to rigorously plan ahead of time in complicated cases. Moreover, as shown in the studies of Prayer et al., 3D reconstruction is feasible and repeatable. They have proven a high correlation with postnatal and postmortem data, observing that the relative size of the diaphragmatic defect remains the same after the second trimester. This might become a new standard for surgery preparations, as with the advances in 3D printing and new regenerative tissue engineering, we might see the apparition of specially made-to-measure patches that can be used to close the diaphragmatic defect. Our team has also started using this method to help with the surgery prep, and we have had good feedback from the surgery teams. Although the dimensions and morphology of the defect were not a key part of this study, we started collecting data and are interested in further looking into this subject in the future.

Among other therapeutic options that can benefit from early diagnosis and staging is the fetoscopic endoluminal tracheal occlusion (FETO). It is an invasive method that requires an experienced team to operate. It requires a fetoscopic approach that aims to implant a detachable balloon during gestation in the trachea of the fetus. The balloon is then removed around 34 weeks. Inclusion criteria for the procedure might vary from center to center, but they usually include singleton pregnancy and isolated CDH with a severe prognosis [31]. Regarding lung size, an increase in volume has been observed in a majority of patients. Ultrasound changes can be visualized as soon as 48 h after the procedure. At the moment, the exact indications and results of the method are still being discussed. While studies show an increase in survivability, they have also proved that there is a high probability of premature birth and other complications, such as premature rupture of membranes. A connection has also been seen between rates of extracorporeal membrane oxygenation (ECMO) utilization and severe pulmonary hypertension. It appears that the risk for these two complications is less in patients that underwent ECMO [32].

An older technique that might still prove its usefulness is the in utero correction of the defect. With newer and more advanced materials and imaging methods, this rather old method might still be improved upon and come forth as a viable alternative to the FETO procedure. It is worth mentioning that there is great debate about which method is more suited to be used in the case of CDH, with both having pros and cons. As the FETO technique is more recent, there is a need for further and rigorous study of the method before it can be considered the standard of therapy in these cases. It is important to take notice of the fact that in utero correction has been proven to be effective only as long as there is no liver ascension in the thoracic cavity. This clearly shows that this method is also not perfect and that it can be improved upon [33,34].

Among the limitations of our study, we have to take into account that not all the patients were monitored in the same hospital and that many of them came from other hospitals for investigations. This makes communication with the clinician difficult, and it also has the added effect of not having access to the full case file of the patients and most importantly to the ultrasound imaging. Another limitation is the fact that there were some cases when the MRI series were suboptimal in terms of quality, but this is to be expected when working with fetuses that present physiological movements during gestation. Finally, we may add the fact that we lack access to the patient files and imaging from other hospitals, thus having a smaller lot of patients in this study. In the future, we hope to recruit more patients for our studies.

## 5. Conclusions

Correct assessment and management of CDH patients can be difficult, and, in some cases, MRI or 3D ultrasound studies might not be easily accessible. In these cases, to not lose time in a somewhat small window, we should look at more accessible 2D techniques that are readily available. From what we have found, the transverse diameter, thoracic area, and thoracic circumference are the most reliable values when it comes to assessing severity. We should also not forget about the LHR, but we advise against using it as the only source of severity assessment. This information allows us to also put a greater value on highly artifacted MRI examinations in which volumetry is not obtainable, as we could also make good use of the available sections. Early diagnosis will allow better management of each patient, as the development of FETO might prove to have great benefits in the future, aiding in the recovery of lost pulmonary volume.

## Figures and Tables

**Figure 1 diagnostics-14-00641-f001:**
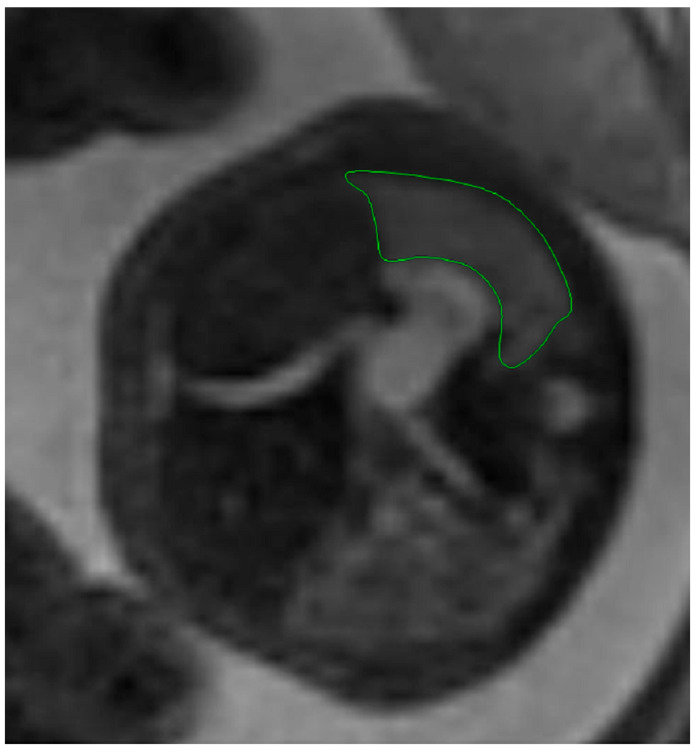
MRI axial T2 weighted image of the fetus showing the tracing method for calculating the lung volume, using the RadiAnt DICOM Viewer program, version number 2022.1.1. The green line represents the lug area on the nonherniated side.

**Figure 2 diagnostics-14-00641-f002:**
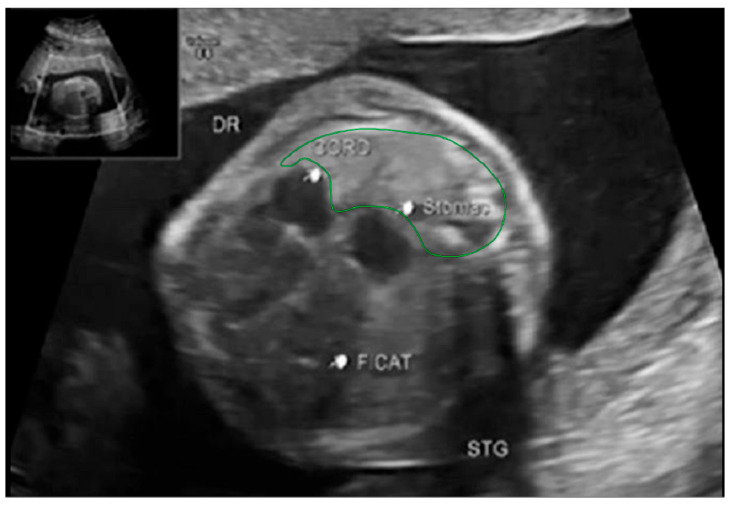
Ultrasound axial four-chamber image of the fetus showing the tracing method for calculating the LHR, using the RadiAnt DICOM Viewer program, version number 2022.1.1. The green line represents the lug area on the nonherniated side.

**Figure 3 diagnostics-14-00641-f003:**
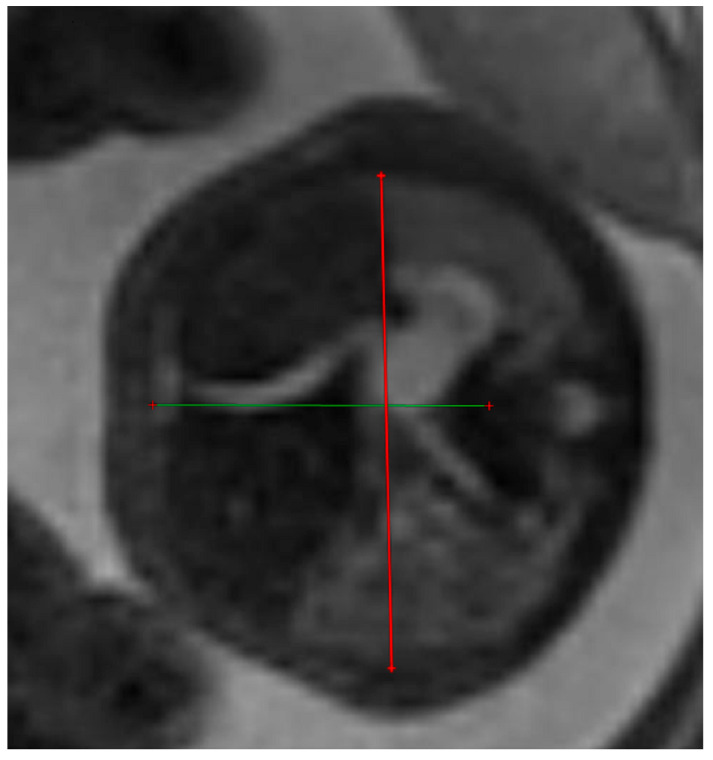
MRI axial T2 weighted image of the fetus showing how to measure the anterior–posterior and transverse diameters of the thorax, using the RadiAnt DICOM Viewer program, version number 2022.1.1. The green lines represent the lug transverse and anterior-posterior diameters.

**Figure 4 diagnostics-14-00641-f004:**
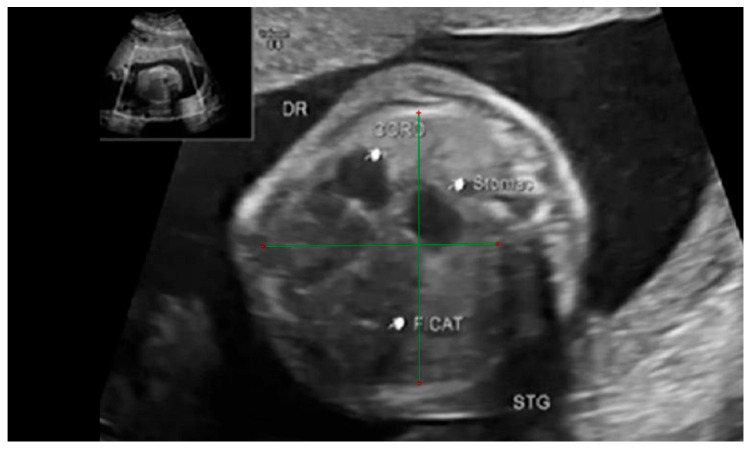
Ultrasound axial four-chamber image of the fetus showing how to measure the anterior–posterior and transverse diameters of the thorax, using the RadiAnt DICOM Viewer program, version number 2022.1.1. The green lines represent the lug transverse and anterior-posterior diameters.

**Figure 5 diagnostics-14-00641-f005:**
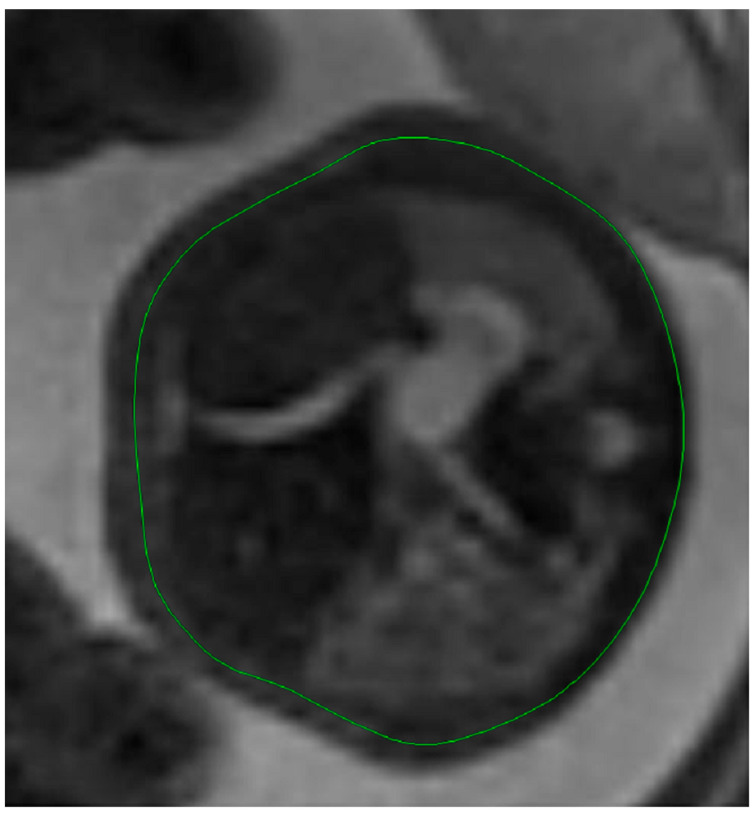
MRI axial T2 weighted image of the fetus showing how to measure the thoracic circumference, using the RadiAnt DICOM Viewer program, version number 2022.1.1. The green line represents the thoracic circumference.

**Figure 6 diagnostics-14-00641-f006:**
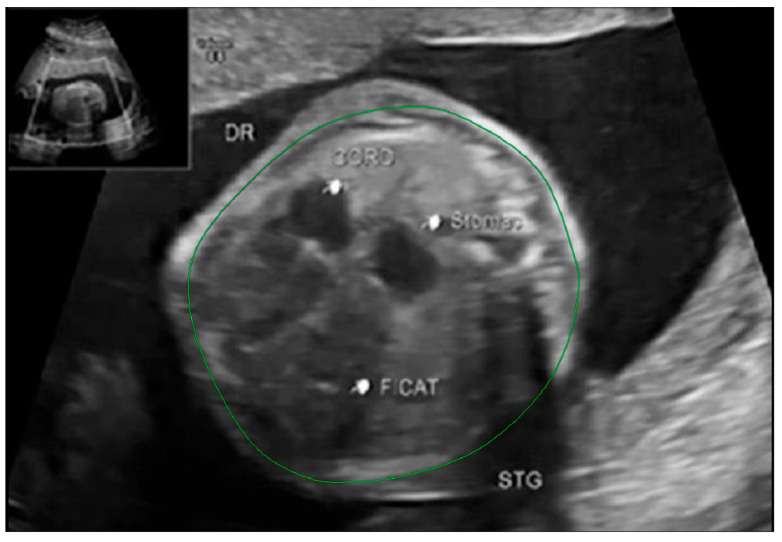
Ultrasound axial four-chamber image of the fetus showing how to measure the thoracic circumference, using the RadiAnt DICOM Viewer program, version number 2022.1.1. The green line represents the thoracic circumference.

**Figure 7 diagnostics-14-00641-f007:**
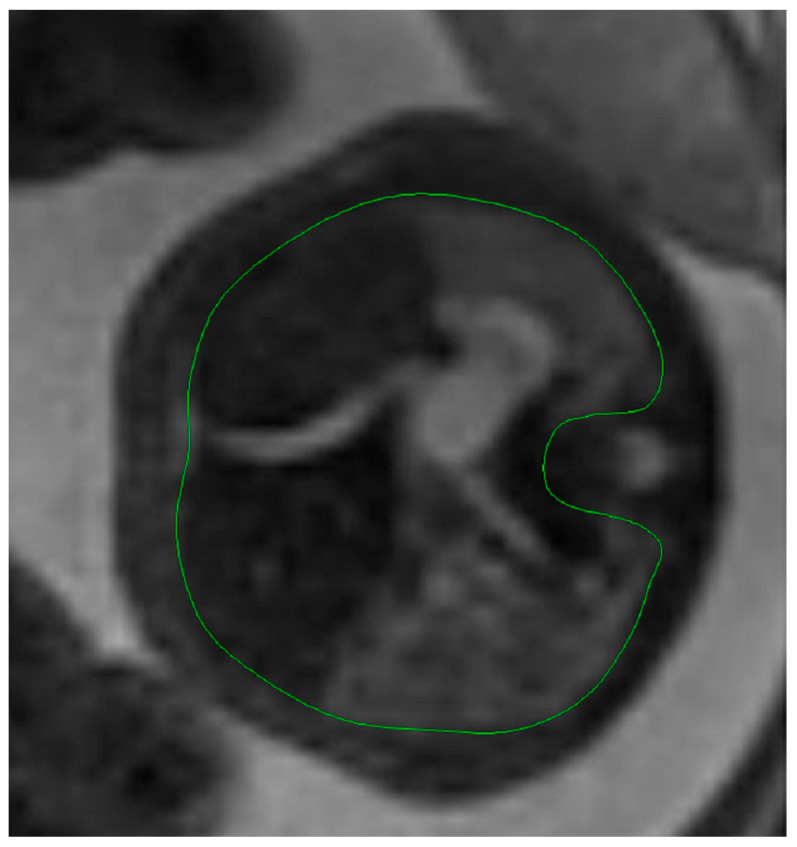
MRI axial T2 weighted image of the fetus showing how to measure the thoracic area, using the RadiAnt DICOM Viewer program, version number 2022.1.1. The green line represents the thoracic area.

**Figure 8 diagnostics-14-00641-f008:**
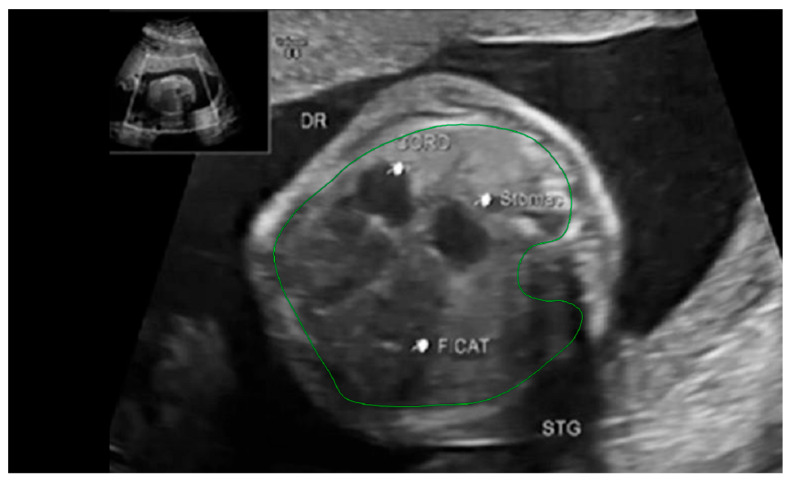
Ultrasound axial four-chamber image of the fetus showing how to measure the thoracic area, using the RadiAnt DICOM Viewer program, version number 2022.1.1. The green line represents the thoracic area.

**Figure 9 diagnostics-14-00641-f009:**
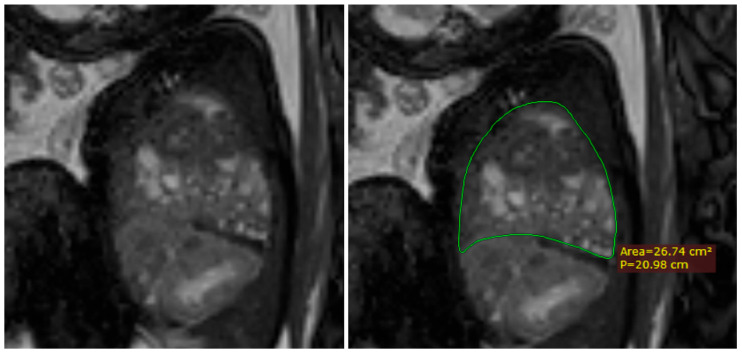
MRI sagittal T2 weighted image of the fetus showing the tracing method for calculating the thoracic volume, using the RadiAnt DICOM Viewer program, version number 2022.1.1. The green line represents the thoracic area on the herniated side. The p in the image stands for perimeter.

**Table 1 diagnostics-14-00641-t001:** Statistical analysis of the distribution of various variables from our study.

Variable	Minimum	Maximum	Mean	Std. Deviation	*p* Anderson–Darling	Interpretation
Observed thoracic tr mm	42.00	77.00	59.21	9.35	0.8132	Gaussian distribution
Observed thoracic ap mm	30.00	65.00	44.00	10.39	0.0556	Gaussian distribution
Observed thoracic circumference mm	147.00	286.00	217.58	40.64	0.6961	Gaussian distribution
Observed thoracic area mm^2^	1219.00	4656.00	2765.53	1037.80	0.0721	Gaussian distribution
Observed thoracic volume mL	31.42	177.79	84.60	50.38	0.0040	non-Gaussian distribution
Observed lung-to-thoracic volume ratio	0.05	0.39	0.18	0.08	0.0407	non-Gaussian distribution
Expected thoracic tr mm	40.80	66.30	54.74	7.95	0.1238	Gaussian distribution
Expected thoracic ap mm	29.40	56.40	43.79	8.83	0.1289	Gaussian distribution
Expected thoracic circumference mm	144.60	254.30	204.09	34.23	0.1457	Gaussian distribution
Expected thoracic area mm^2^	1565.00	4599.00	3070.32	1010.04	0.0578	Gaussian distribution
Expected total pulmonary volume (mL) VOCAL study	13.54	64.67	38.45	16.99	0.0589	Gaussian distribution
Expected thoracic volume	27.10	114.23	70.22	28.75	0.0761	Gaussian distribution
Total pulmonary volume index VOCAL comparison	0.08	0.69	0.37	0.14	0.2561	Gaussian distribution
Expected lung-to-thoracic volume ratio	0.50	0.57	0.54	0.02	0.1867	Gaussian distribution
Observed total pulmonary volume (mL)	1.6800	31.4700	14.0316	7.3288	0.9550	Gaussian distribution
Expected total pulmonary volume (mL) mean MRI studies	15.2444	64.0786	39.7768	15.2639	0.1321	Gaussian distribution
Total pulmonary volume index MRI comparison	0.0709	0.6281	0.3494	0.1344	0.4444	Gaussian distribution

**Table 2 diagnostics-14-00641-t002:** Statistical analysis of the correlation between various variables from our study.

Variable 1	Variable 2	*p* Student	Significance
Observed thoracic tr mm	Expected thoracic tr mm	0.0002	HS
Observed thoracic ap mm	Expected thoracic ap mm	0.8797	NS
Observed thoracic circumference mm	Expected thoracic circumference mm	0.0005	HS
Observed thoracic area mm^2^	Expected thoracic area mm^2^	0.0042	S
Observed thoracic volume mL	Expected thoracic volume	0.0518	NS
Observed lung-to-thoracic volume ratio	Expected lung-to-thoracic volume ratio	0.0000	HS
Observed total pulmonary volume (mL)	Expected total pulmonary volume (mL) mean MRI studies	0.0000	HS
Observed total pulmonary volume (mL)	Expected total pulmonary volume (mL) VOCAL study	0.0000	HS
Total pulmonary volume index MRI comparison	Total pulmonary volume index VOCAL comparison	0.0035	S
Expected total pulmonary volume (mL) mean MRI studies	Expected total pulmonary volume (mL) VOCAL study	0.0112	S

## Data Availability

The data presented in this study are available on request from the corresponding author.

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
