# Peer review of "Thoracic Biometry in Patients with Congenital Diaphragmatic Hernia, a Magnetic Resonance Imaging Study"

_diagnostics, 2024, doi:10.3390/diagnostics14060641_

Round 1

Reviewer 1 Report

Comments and Suggestions for Authors

There are several issues that the authors require to improve.

1.  In line 36, the number 50.000 is the 50 thousands or 50? It is a little confusing for this value.

2. The authors only mentioned that their study was approved ethically, please also provide the protocol number.

3. The descriptions for Fig 1 to 5 is not clear in the main text, and citations of the those figures cannot be found. Also, those images were all from MRI? Are there any corresponding ultrasound images?

4. In Fig 1, 3,4, 5, what is the P standing for in the images?

5. For results, the authors may provide some comparison with ultrasound images?

Author Response

Good day,

Thank you for the insightful review of our article.

As per your comments, cleared the problem with the value in line 36.

We added the protocol number of the ethics committee decision.

Finally, we changed the images in the article to make it clearer how the measurements should be made. Additionally, we added ultrasound images for a comparison between the two methods, as you suggested. Lastly, we improved the descriptions of the images and also added citations in the main text.

To address the comparison between ultrasound and MRI images, unfortunately, we do not have access to all the ultrasound imaging of our patients, as many of them were recommended to us from other hospitals, thus making a thorough comparison impossible at this time.

Thank you for your advice.

Reviewer 2 Report

Comments and Suggestions for Authors

This is a multicenter retrospective study on the biometric measurements using MRI examinations in congenital diaphragmatic hernia (CDH). The authors investigated that correlation between 2D and 3D volume biometric values obtained by MRI and ultrasound and demonstrated that the transverse diameter, thoracic area, and thoracic circumference are the most reliable values when it comes to assessing severity. This study may provide some useful information on the usefulness of MRI follow-up in CDH cases. I have some comments.

<Comments>

1. In line 384, What’s the “VOCAL” study? Please add the references.

2. In line 84, Please describe the full term of the abbreviation, “MRI.

3. In line 384, What’s the “LHR method”? Please describe the full term of the abbreviation, “LHR and add the references.

4. It would be better to briefly describe the Introduction section.

5. Please add the limitation of this study in the Discussion section.

Comments on the Quality of English Language

Moderate editing of English language required.

Author Response

Good day,

Thank you for the insightful review of our article.

As per your comments, we have added references and described the full terms of the VOCAL,

MRI and LHR acronyms at the 71, 72, and 80 lines.

We have also reduced the introduction to make it clearer.

Finally, we added a paragraph where we discuss about the limitations of the study.

Thank you for your advice.

Round 2

Reviewer 1 Report

Comments and Suggestions for Authors

no further comments

Reviewer 2 Report

Comments and Suggestions for Authors

The revised version was improved with well-prepared response to most comments raised by reviewers.
Although the revision for papers is not enough for my comments, the authors put much effort to revise manuscripts.
Thank you for your effort.